# The Association between Post-Traumatic Stress Disorder and Psychological Distress among Primary School and Middle School Teachers during the COVID-19 Epidemic: A Moderated Mediation Analysis

**Jia-Xin Liang [1], Ying Gao [1], I-Hua Chen [2]🆔, Xiu-Mei Chen [3],*🆔 and Yuan-Yuan Zheng [4],***

[1] School of Education Science, Minnan Normal University, Zhangzhou 363000, China
[2] Chinese Academy of Education Big Data, Qufu Normal University, Qufu 273100, China
[3] Faculty of Education, Qufu Normal University, Qufu 273100, China
[4] International College, Krirk University, Bangkok 10220, Thailand
* Correspondence: cxm@sdyu.edu.cn (X.-M.C.); users0204@gmail.com (Y.-Y.Z.)

**Abstract:** Background: The outbreak of the coronavirus disease (COVID-19) and its rapid spread may have led to individuals developing post-traumatic stress disorder (PTSD) and psychological distress. Under this context, teachers merit more attention as a group with high levels of work stress. The purpose of this study was to verify the relationship between PTSD and psychological distress and to explore sleep problems as a possible mediator in the relationship between PTSD and psychological distress, as well as the moderator of internet gaming disorders (IGD) in the relationship between sleep problems and psychological distress. Methods: A total of 11,014 Chinese primary and middle school teachers participated in this study. The survey was conducted online between 25 May and 30 June 2020. Results: PTSD was shown to have both a direct and indirect effect on teachers' psychological distress. The indirect effect was mediated by sleep problems. IGD played a moderating role between sleep problems and psychological distress. Conclusions: During the COVID-19 pandemic, PTSD has been shown to have had a serious impact on the psychological stress of teachers, which was mediated by sleep problems. In addition, IGD raised the harm brought from sleep problems on teachers' mental health.

**Keywords:** post-traumatic stress disorder; psychological distress; sleep problems; internet gaming disorder; COVID-19

## 1. Introduction

### 1.1. The Impact of COVID-19 on Teachers

The outbreak of the coronavirus disease (COVID-19) had many physical and psychological effects on people. With the large-scale spread of the pandemic, much attention has been given to the state of peoples' mental health [1–4]. Some of these studies have been done in consideration of post-traumatic stress disorder (PTSD) [4,5]. Studies have shown that the speed of the pandemic's spread, home isolation, contact with suspected infected individuals, the pain of bereavement, the stress of unemployment, and the uncertainty of the future can all cause people to feel miserable and make them more prone to PTSD [4,6]. PTSD is characterized by persistent and intrusive memories of traumatic events, high vigilance, avoidance of trauma-related cues, and negative changes in thinking and emotions [7]. PTSD causes harm to peoples' mental health. For example, studies have found that people with traumatic experiences, such as experiencing Severe Acute Respiratory Syndrome (SARS), the Ebola virus, earthquakes, or hurricanes, are more likely to experience psychological distress [8–11]. After the 1999 Marmara earthquake in Turkey and Hurricane Mitch in Nicaragua, studies found that people had high levels of comorbid

PTSD and depression (i.e., 67.5% and 79% for the PTSD and depression comorbid rates in both traumatic experiences) [10,11].

In addition to the impact of PTSD on psychological distress, sleep issues can be another variable that may contribute to psychological distress. During the outbreak of this pandemic, many destabilizing social effects led to increased sleep problems [12,13], and some studies have investigated the association between such sleep problems and psychological distress [12,14,15]. The results indicated that individuals with sleep problems were more likely to develop depression and anxiety than individuals with healthy sleep [12,14,15].

In the context of COVID-19, studies have investigated both the effects of PTSD and sleep problems on psychological distress, but no studies have taken all three details into account, which could lead to doubt regarding the effects of these two variables on mental illness within the pandemic context. This is because the diagnostic indicators of PTSD include items reflecting sleep problems [15], making it hard to avoid the confounding factor of sleep problems as part of the impact of PTSD on peoples' mental health unless both PTSD and sleep problems can be included in the same analysis model. Therefore, it is necessary to discuss these three variables at the same time, so that the influence of each variable on mental illness can be more accurately seen.

On the topic of mental health during the pandemic, internet gaming disorder (IGD) has also received considerable attention. IGD is an excessive and prolonged pattern of internet gaming that results in a range of cognitive and behavioral symptoms, including a gradual loss of control over gaming, tolerance, and withdrawal symptoms similar to those of substance use disorder [7]. The basic characteristic is constant and regular participation in computer games, especially group games, for many hours at a time [7]. It is reported that greater psychological distress increases the amount of time primary school students spend playing games at home [16]. In a recent survey, 23.7% of primary school students reported that their time spent playing games at home has increased, and the results show that depression, anxiety, and stress are significantly associated with IGD [16]. In another survey conducted during the pandemic, IGD was proven to be a strong predictor of psychological distress, and higher levels of IGD were associated with more depression, anxiety, and stress [17–19]. Among these results, IGD was treated as a dependent or an explanatory variable [16,19]. However, the association between IGD and psychological distress may not only exist in the direct correlation between the two, but IGD may also combine with other variables to jointly influence psychological distress. For example, some scholars have suggested that internet games can easily interfere with sleep [18]. As such, whether IGD exacerbates the effects of sleep problems on psychological distress requires further exploration.

Many studies have been done on the mental health status of various groups during the pandemic, including frontline medical staff, adolescents, and college students [1,20,21]. However, few studies have looked at the psychological status of primary and middle school teachers. After the outbreak of COVID-19 in December 2019, China immediately took a series of countermeasures, which included the closure of public entertainment places and educational institutions [22] and moving to adopt online teaching. Teaching in the virtual environment requires teachers to be proficient in different remote teaching tools and to change their teaching strategies, which undoubtedly poses a huge challenge to teachers [23–25]. Moreover, it has been shown that teachers experience increased pressure when they must use unfamiliar technologies to conduct online teaching [26]. Therefore, it is pertinent to conduct a large-scale investigation into the psychological distress of such a high-risk demographic.

### 1.2. Research Purpose and Hypothesis

During the pandemic's outbreak, teachers who were unprepared to adapt to these new circumstances were required to adopt online teaching methods which may have made them vulnerable to psychological distress [23,27–29]. As such, this study focused on this demographic to investigate the possible factors influencing their psychological distress. To

fully explain the underlying mechanism, a conceptual model was proposed (Figure 1). As shown in this model, we examined the association among teachers' PTSD (i.e., teachers at high risk of PTSD and normal individuals), psychological distress, sleep problems, and IGD. What follows are the relevant empirical works that have led us to hypothesize the model pathway.

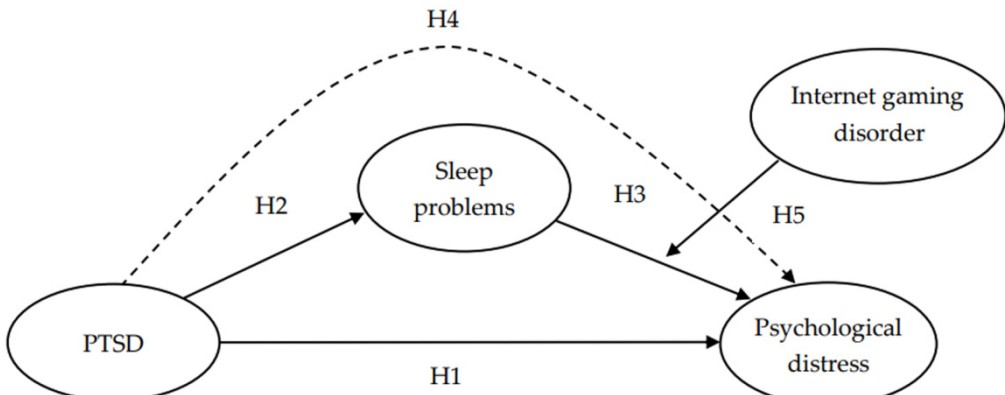

**Figure 1.** Research model and hypotheses. H4 is the mediating effect of sleep problems. (PTSD (Post-traumatic stress disorder) → Sleep problems → Psychological distress). The dashed line below H4 represents the indirect effect between PTSD and Psychological distress.

First, studies have shown that PTSD is significantly related with depression. One study from the United States showed that 24.2% of children with traumatic experiences showed depressive symptoms, and adolescents who had experienced a traumatic event were 2.6 times more likely to develop depression than those who had not [30]. In addition, studies have shown that PTSD is positively correlated with depression, anxiety, and stress [31]. Therefore, we hypothesized that PTSD is positively correlated with psychological distress (H1).

Numerous studies have shown that sleep problems are closely related to PTSD, and studies have shown that more than 50% of individuals with PTSD will report sleeping problems [32,33]. Sleep problems are also associated with depression, stress, anxiety, and other factors [12]. According to the 3P (predisposing, precipitating, and perpetuating factors) model of insomnia as proposed by Spielman and Glovinsky, sleep problems lead to an increase in peoples' existing stress levels [34]. Therefore, the following hypotheses were proposed:

**H2.** *PTSD is positively correlated with sleep problems.*

**H3.** *Sleep problems are positively correlated with psychological distress.*

Given that both PTSD and sleep problems are predictors of teachers' psychological distress and that PTSD may also influence sleep problems, it is reasonable to hypothesize that sleep problems mediate the relationship between PTSD and psychological distress (H4).

Studies indicate that psychological distress is closely related to the sleep status of individuals, and depressed individuals in particular will generally have a shorter reported sleep time [35,36]. In practice, however, not all people with sleep problems are equally troubled. Furthermore, individuals with the same level of sleep problems can also have different levels of mental health problems (such as anxiety levels, depression levels, stress levels), so there may be other important moderators between sleep problems and psychological distress. According to Davidson's emotional resilience theory, some individuals may produce positive or negative emotions too easily, which is detrimental to their physical and mental health [37]. The emotional experience of online gaming tends to be a roller coaster ride depending on whether the player is winning or losing in the game. During the COVID-19 pandemic, teachers not only have had to bear heavy schoolwork pressures, but they have also had to manage the general psychological pressures brought by the pan-

demic. With all the stress that has come due to all the changes, sleep for many has suffered. According to the self-regulation self-depletion model, when an individual's self-regulation resources are insufficient and the individual is in a state of self-depletion, this will lead to a decline in the individual's self-control, leading to impulses that are not conducive to the individual's wellbeing that appear in order to alleviate the depletion that the individual is suffering [38]. Insomnia is a typical manifestation of ego depletion [39,40]. Therefore, when an individual suffers from sleep problems, an individual with IGD may have positive or negative emotions brought about by internet or gaming addictions, aggravating the impact of sleep problems on their psychological distress. Therefore, this study hypothesized that online game addiction plays a moderating role in the effect of sleep problems on psychological distress (H5).

## 2. Materials and Methods

### 2.1. Participants and Procedure

An online survey was conducted using teachers from Jiangxi, Sichuan, and Shandong between 25 May and 30 June 2020, during the COVID-19 pandemic. The study was approved by the ethics committee of the Jiangxi Psychological Consultant Association (IRB ref: JXSXL-2020-J013). Informed consent was obtained from all individual participants included in the study. There were 11,014 teachers who participated in the investigation, with 3157 (28.7%) males and 7857 (71.3%) females. All teachers were from primary or middle schools (primary school teachers = 6921, 62.8%; middle school teachers = 4093, 37.2%). Among them, 10,566 (95.9%) worked at public schools and 448 (4.1%) at private schools. The range of teaching experience was from 1 year to 5 years (1 year = 2925, 26.6%; 2 years = 1888, 17.1%; 3 years = 1445, 13.1%; 4 years = 1187, 10.8%; 5 years = 3569, 32.4%).

### 2.2. Measures

In this study, the Chinese PTSD Checklist for Diagnostic and Statistical Manual of Mental Disorders (DSM-5) (PCL-5) was used to evaluate teachers' PTSD in order to understand the teachers' situation during the COVID-19 pandemic. The 21-item Depression, Anxiety, and Stress Scale (DASS-21) was used to assess teachers' psychological distress; the 9-item Internet Gaming Disorder Scales-Short Form (IGDS-SF9) was used to assess the extent of teachers' IGD; one item was used to assess teachers' sleep problems. These instruments are described in detail below.

#### 2.2.1. The Chinese Post-Traumatic Stress Disorder Checklist for DSM-5 (PCL-5)

The PCL-5 is one of the most widely used self-report measures in the DSM-5 for the assessment of PTSD [41]. The PCL-5 includes 20 items which are rated using a five-point Likert scale ranging from 0 (Not at all) to 4 (Extremely). In this study, teachers were divided into a high-risk PTSD group (score higher than 33) and a normal group according to the diagnostic criteria of DSM-5 [42,43] and based on the total score on the PCL-5. The Cronbach's $\alpha$ of the PCL-5 in this study was 0.95 for both groups of schoolteachers.

#### 2.2.2. The Depression, Anxiety, Stress Scale-21 (DASS-21)

The DASS-21 is a shortened version of the Depression and Anxiety Stress Scale-42 developed by Lovibond and Lovibond [44]. It consists of three subscales that assess the extent of one's depression, anxiety, and stress. Seven items on each subscale are scored using a four-point Likert scale ranging from 0 (Did not apply to me at all) to 3 (Applied to me very much, or most of the time). Sample items are "I felt that life was meaningless" (depression); "I was aware of the action of my heart, without any physical exertion" (anxiety); and "I found it hard to wind down" (stress) [45]. The Cronbach's $\alpha$ of the three subscales in the current study were as follows: (i) the depression scale was 0.92 for both categories of schoolteachers; (ii) the anxiety scale was 0.91 for both categories of schoolteachers; (iii) the stress scale was 0.90 for primary school teachers and 0.91 for middle school teachers. To judge whether the participating schoolteachers had any mental illness, the DASS-21 scores

were multiplied by two and assessed using the following cutoff: (i) depression: less than 9 points is normal status, more than 10 is mental illness; (ii) anxiety: less than 7 points is normal status, more than 8 is mental illness; (iii) stress: less than 14 points is normal status, more than 15 is mental illness [46].

### 2.2.3. Sleep Problems

We used a single item to measure sleep problems, taken from the Chinese version of the Fear of COVID-19 Scale [47]: "I couldn't sleep well because I was worried about getting COVID-19." This measure was rated using a five-point scale ranging from 1 (Totally disagree) to 5 (Completely agree). The severity of the sleep problems was indicated by the item score, with higher scores indicating a higher level of sleep problems.

### 2.2.4. The Nine-Item Internet Gaming Disorder Scales-Short Form (IGDS-SF9)

The IGDS-SF9 is a short psychometric tool for the assessment of IGD which was developed by Pontes and Griffiths [48]. The IGDS-SF9 is a self-report instrument with items rated using a five-point Likert scale, ranging from 1 (Never) to 5 (Very often). After totaling all items to obtain one overall score, a higher score indicates a higher level of internet gaming disorder. A sample item is, "Do you systematically fail when trying to control or cease your gaming activity?" The IGDS-SF9 has a well-established unidimensional structure, and the Cronbach's $\alpha$ in this study was 0.95 for both categories of schoolteachers.

### 2.3. Data Analyses

In this study, SPSS 21 was used first to deal with descriptive statistics, and Pearson's correlation was used to explore correlations between variables. Next, we tested the research hypotheses about the mediating effect of sleep problems with Partial Least Squares Structural Equation Modeling (PLS-SEM) [49]. PLS-SEM can be used to examine data without distribution assumptions [50]. PLS can be divided into an outer model and inner model; the outer model can reflect the loading or weight of each indicator, showing the data's relationship with the model, and can be used to evaluate the construct validity of the model [50]. The outer model contains a formative indicator and a reactive indicator. All indicators used in this study were reactive indicators. The model was evaluated using the internal consistency reliability (Cronbach's $\alpha$, threshold $\geq 0.7$), composite reliability, convergent validity (AVE, threshold $\geq 0.5$), and discriminant validity (HTMT, threshold $\leq 0.85$) [49,51].

The linear relationship can be seen from the inner model (also called structural model) between constructs through the path coefficient (range from $-1$ to $+1$) and can be used to evaluate whether there is a significant difference between indicators by comparing it with zero. In this study, the hypotheses were tested with bootstrapping ($n = 5000$) and 95% confidence intervals (CI) [52,53]. We selected the coefficient of determination ($R^2$, with 0.67, 0.33, and 0.19 displaying substantial, moderate, or weak levels, respectively), and effect size ($f^2$, with 0.35, 0.15, and 0.02 displaying large, medium, or weak effects, respectively) to evaluate the endogenous constructs [54]. Given that there has been no common agreement of model fit criterion in PLS-SEM and that the SmartPLS manual recommends that researchers should be very cautious to report and use model fit in PLS-SEM [51,55], we did not report any indicators related to model fitting (e.g., SRMR, d_ULS, d_G, Chi-square, NFI). Finally, the mediating moderation model hypothesized in this study was also tested by PLS. PLS can be used to draw simple slope analyses of $\pm$ one *SD*.

## 3. Results

### 3.1. Descriptive Statistics and Pearson's Correlations

Table 1 shows the descriptive statistics including the means (standard deviations; *SD*) and Pearson's correlations (*r*) of each variable. According to the diagnostic criteria of the DSM-5 [42,43], a total of 978 participants were classified as the group with a high risk for PTSD, while the remaining participants made up the normal group ($n = 10,036$). As for psychological distress, 3100 suffered from psychological distress (28.14%), including

2901 (26.34%) having anxiety, 2273 (20.63%) having depression, and 1114 (10.11%) having stress. As expected, all variables showed a significant positive correlation with each variable ($p < 0.01$), among which the lowest correlation was with IGD and sleep problems ($r = 0.328$, $p < 0.01$), while PTSD had the highest correlation with psychological distress ($r = 0.610$, $p < 0.01$). The relationship between the variables supports the follow-up test of the research hypotheses.

**Table 1.** Descriptive statistics and Pearson's correlations of each study variable.

|  | **M** | *SD* | **1** | **2** | **3** | **4** |
|---|---|---|---|---|---|---|
| 1. PTSD | 13.139 | 12.090 | 1 | | | |
| 2. SP | 1.82 | 0.692 | 0.402 ** | 1 | | |
| 3. PD | 15.091 | 20.191 | 0.610 ** | 0.419 ** | 1 | |
| 4. IGD | 13.520 | 6.048 | 0.448 ** | 0.328 ** | 0.486 ** | 1 |

Notes. PTSD = Post-traumatic stress disorder; SP = Sleep problems; PD = Psychological distress; IGD = Internet gaming disorder. ** $p < 0.01$; PD is calculated by multiplying the DASS-21 score by 2.

### 3.2. Assessment of Reflective Outer Model

To test the relationship between PTSD, sleep problems, psychological distress, and IGD, the hypothetical model was established, and PLS was used to test it. Table 2 shows the outer model results, with the discriminant validity, composite reliability, Cronbach's $\alpha$, and AVE values meeting the recommended criteria [49,56–58]. The HTMT values of all indicators, except when compared to themselves, were ≤0.85, indicating that the discriminant validity of each indicator was good. In addition, the composite reliability and Cronbach's $\alpha$ values were both more than the threshold value of 0.7, indicating that the model had good reliability.

**Table 2.** Assessment of the measurement model.

|  | **HTMT** | | | | **CR** | **Cronbach's $\alpha$** | **AVE** |
|---|---|---|---|---|---|---|---|
|  | **1** | **2** | **3** | **4** | | | |
| 1. PTSD | 1.00 | | | | | Not applicable | |
| 2. SP | 0.248 | 1.00 | | | | Not applicable | |
| 3. PD | 0.419 | 0.419 | 0.789 | | 0.972 | 0.969 | 0.623 |
| 4. IGD | 0.333 | 0.330 | 0.492 | 0.855 | 0.961 | 0.954 | 0.731 |

Notes. HTMT = Heterotrait-Monotrait Ratio; CR = Composite Reliability; Cronbach's $\alpha$ = Cronbach's Alpha; AVE = Average Variance Extracted.

### 3.3. Assessment of Inner Model

Tables 3–5 and Figure 2 show the bootstrapping results for the relationships between anxiety, stress, and depression. The results reveal that PTSD was significantly and positively correlated with psychological distress. Specifically, PTSD positively predicted anxiety ($f^2 = 0.201$), stress ($f^2 = 0.180$), and depression ($f^2 = 0.209$) with medium effects (H1). It also had a positive prediction on sleep problems ($f^2 = 0.248$) with medium effects (H2). Similarly, sleep problems also positively predicted anxiety ($f^2 = 0.212$), stress ($f^2 = 0.203$), and depression ($f^2 = 0.180$; H3). As for H4, when sleep problems were included as a mediator, the path coefficient for anxiety changed from 0.409 to 0.325, the path coefficient for stress changed from 0.390 to 0.308, and the path coefficient for depression changed from 0.416 to 0.340, all of which show that there were complementary and partial mediating effects in these three dimensions [51,59]. The $R^2$ values for the endogenous variables of anxiety, stress, and depression were 0.274 (small), 0.256 (small), and 0.261 (small), respectively. The average $R^2$ was 0.264.

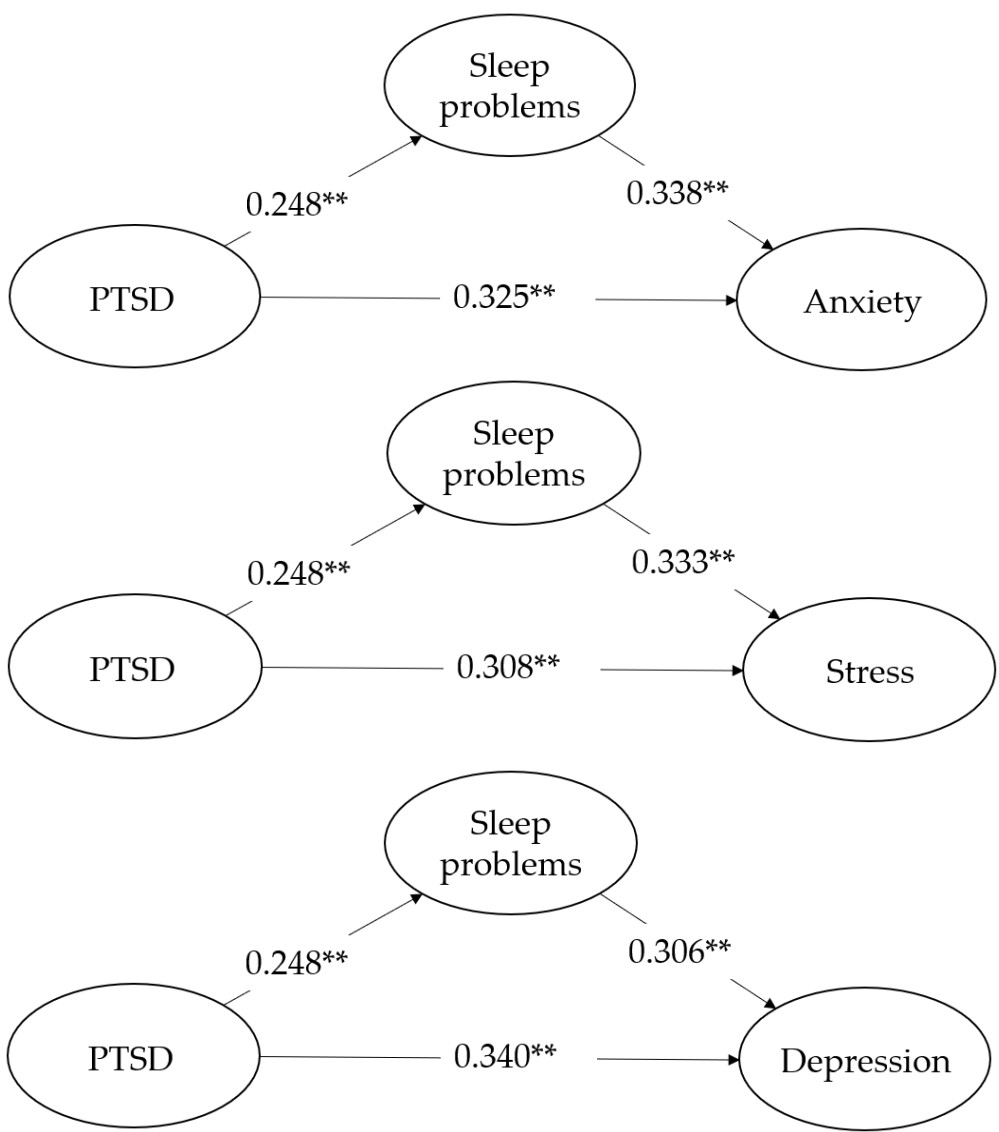

**Figure 2.** The PLS-SEM model with path coefficients and factor loading. Moderated mediation of IGD (Internet games disorder) is not included in the above models; ** *p* < 0.01. PTSD represents Post-traumatic stress disorder.

**Table 3.** Hypotheses testing for inner model results of anxiety.

| Hypotheses | Effect | | SD | t | p | 95% CI |
|---|---|---|---|---|---|---|
| (H1) PTSD → Anxiety | Path Coeff. | 0.409 | 0.012 | 33.408 | <0.01 | [0.385, 0.433] |
| (H2) PTSD → SP | Path Coeff. | 0.248 | 0.011 | 23.088 | <0.01 | [0.226, 0.269] |
| (H3) SP → Anxiety | Path Coeff. | 0.418 | 0.010 | 42.894 | <0.01 | [0.399, 0.437] |
| (H4) PTSD → SP → Anxiety | Path Coeff. | 0.084 | 0.004 | 19.513 | <0.01 | [0.075, 0.092] |
| (H5) SP*IGD → Anxiety | Path Coeff. | 0.057 | 0.012 | 4.889 | <0.01 | [0.034, 0.080] |

Notes. The *t*-value should be greater than 1.96 (plus or minus) and the *p*-value should be less than 0.05 to be considered significant. The interval of 95% Confidence Intervals (CI) cannot contain zero. *SD* = Standard Deviation (STDEV).

**Table 4.** Hypotheses testing for inner model results of stress.

| Hypotheses | Effect | | SD | t | p | 95% CI |
|---|---|---|---|---|---|---|
| (H1) PTSD → Stress | Path Coeff. | 0.390 | 0.012 | 33.562 | <0.01 | [0.367, 0.413] |
| (H2) PTSD → SP | Path Coeff. | 0.248 | 0.011 | 23.088 | <0.01 | [0.226, 0.269] |
| (H3) SP → Stress | Path Coeff. | 0.410 | 0.010 | 43.130 | <0.01 | [0.392, 0.429] |
| (H4) PTSD → SP → Stress | Path Coeff. | 0.082 | 0.004 | 19.470 | <0.01 | [0.074, 0.091] |
| (H5) SP*IGD → Stress | Path Coeff. | 0.034 | 0.010 | 3.346 | <0.01 | [0.014, 0.054] |

Notes. The *t*-value should be greater than 1.96 (plus or minus) and the *p*-value should be less than 0.05 to be considered significant. The interval of 95% Confidence Intervals (CI) cannot contain zero. *SD* = Standard Deviation (STDEV).

**Table 5.** Hypotheses testing for inner model results of depression.

| Hypotheses | Effect | | SD | t | p | 95% CI |
|---|---|---|---|---|---|---|
| (H1) PTSD → Depression | Path Coeff. | 0.416 | 0.012 | 34.226 | <0.01 | [0.392, 0.440] |
| (H2) PTSD → SP | Path Coeff. | 0.248 | 0.011 | 23.088 | <0.01 | [0.226, 0.269] |
| (H3) SP → Depression | Path Coeff. | 0.391 | 0.010 | 39.379 | <0.01 | [0.372, 0.410] |
| (H4) PTSD → SP → Depression | Path Coeff. | 0.076 | 0.004 | 18.530 | <0.01 | [0.068, 0.084] |
| (H5) SP*IGD → Depression | Path Coeff. | 0.058 | 0.011 | 5.093 | <0.01 | [0.035, 0.080] |

Notes. The *t*-value should be greater than 1.96 (plus or minus) and the *p*-value should be less than 0.05 to be considered significant. The interval of 95% Confidence Intervals (CI) cannot contain zero. *SD* = Standard Deviation (STDEV).

In terms of moderation (H5), the results indicate that the moderating effects on anxiety (Path Coeff. = 0.057, *t* = 4.889, *p* < 0.01), stress (Path Coeff. = 0.034, *t* = 3.346, *p* < 0.01), and depression (Path Coeff. = 0.058, *t* = 5.093, *p* < 0.01) were significant (see Tables 3–5).

## 4. Discussion

As a public health emergency, the COVID-19 pandemic has had a huge impact on teachers. This study focused on teachers' psychological and physiological conditions and constructed a moderated medication model which proves that the mental health of teachers with PTSD will be affected by sleep problems and IGD.

In support of our first hypothesis, the path coefficient was significantly and positively correlated with PTSD and psychological distress. This result is the same as in previous studies looking at the relationship between PTSD and psychological distress [31]. Teachers are a group who can face high amounts of pressures [60,61], with regular exposure to students and parents even during the COVID-19 pandemic. In this context in particular, these teachers were under pressure not only when teaching their students, but also as they dealt with the psychological strain of the epidemic [62]. As a psychological disorder, PTSD is related to environmental stressors, which can lead to individuals becoming even more sensitive to psychological distress in the context of the COVID-19 pandemic [63].

Our results also proved Hypotheses 2 and 3 in that PTSD was also significantly and positively correlated with sleep problems, and with sleep problems positively correlated with psychological distress. One interesting finding is that sleep problems were most likely to cause anxiety, followed then by stress and depression in the three dimensions of psychological problems. As one of the most common negative psychological states, the first thing that sleep problems cause teachers is a sense of psychological irritability, and this feeling is the source of anxiety [64]. Many clinical studies have shown that sleep problems can predict the likelihood of developing future psychological distress, particularly anxiety [14,65,66]. For example, alterations in sleep may exacerbate generalized anxiety disorder [65]. As for depression, nighttime insomnia symptoms have also been shown to overlap with depression and anxiety symptoms [67]. Our research results support the

conclusion that PTSD can affect peoples' sleep problems, and that sleep problems can then lead to psychological distress.

The mediating effect of sleep problems in relation to PTSD and psychological distress was also examined (H4). PTSD is related to traumatic events. If someone died from COVID-19, the risk of their family members or relatives suffering from PTSD is greatly increased [4]. Similarly, the treatment of PTSD is often undermined by fragmented rapid eye movement (REM) sleep [68,69], so sleep problems and PTSD are closely related and move in the same direction. Poorer sleep is strongly associated with poorer psychological health [70–72]. Therefore, it could be suggested that when a teacher suffers from PTSD, the kinds of stresses that are placed on them are magnified, possibly affecting their sleep and, in turn, the teacher's psychological health.

Hypothesis 5, that IGD plays a moderating role in the relationship between sleep problems and psychological distress, was supported. These results were statistically significant, although the effect was not large. Regarding the relatively small moderating effect of IGD, it may be because most teachers have been very busy throughout the crisis [73–75] and didn't have much time to play games. However, for the few teachers with excessive internet gaming disorder, sleep problems are more damaging to their mental health. It appears that problematic internet behaviors among teachers should not be overlooked, even if their proportion is not high, since they can still negatively impact their psychological wellbeing over the long term.

## 5. Conclusions

This study found that during the COVID-19 pandemic, teachers, as a group, experiencing high psychological distress were more likely to report PTSD, and this PTSD may have caused teachers to experience increased depression, anxiety, and stress due to an increase in sleep problems. Experiencing severe sleep problems or stressful events increases teachers' likelihood to suffer from IGD, as does the internet becoming their primary means of being social with others or an accessible way to alleviate the pressures of the pandemic. In public health emergencies, teachers' mental health is of extreme importance given their role in society and involvement with younger generations. As such, further exploration is needed to understand and safeguard the mental health status of teachers.

## 6. Limitations

This study demonstrated the mediating effect of sleeping problems between PTSD and psychological distress, as well as the moderating effect of internet games distress between sleeping problems and psychological distress. There are several limitations of this study. The primary limitation of this study is the measure of PTSD. In line with other studies [76–78], we used the PCL-5 to measure symptoms of PTSD. However, some of the items in the PCL-5 measured the emotion of people exposed to trauma, while the negative emotions were also the essence in DASS-21, which was used as the indicator of psychological distress in this study. The overlap between PCL-5 and DASS-21 may affect the accurate association between PTSD and psychological distress, although the discriminant validity of these two latent variables was supported (see Table A1). Even though the measure of PCL-5 is a proxy variable of traumatic experience, future research could still consider reinvestigating the relationship between PTSD and psychological distress by including the data of pure traumatic experiences. Second, only one item was asked about sleep problems, so the measurement of sleep problems may be biased. Third, as the context of the COVID-19 pandemic is particularly unique, one must be cautious in terms of the findings' applicability, as the findings might not be transferable to normal situations. Therefore, future research should explore the mental health of teachers in a daily life context. Fourth, our data collection relied on internet and self-reported data gathering, and this could lead to bias. To avoid this situation, we standardized the data before analysis. Fifth, similar to the second limitation, the subjects of this study were mainly primary and middle school teachers. This model may not be applicable for high school or university teachers.

**Author Contributions:** Conceptualization, Y.G. and J.-X.L.; methodology, I.-H.C. and X.-M.C.; investigation, I.-H.C. and Y.-Y.Z.; data curation, X.-M.C. and I.-H.C.; writing—original draft preparation, J.-X.L. and Y.G.; writing—review and editing, I.-H.C. and X.-M.C.; visualization, Y.-Y.Z.; supervision, I.-H.C. and Y.-Y.Z. All authors have read and agreed to the published version of the manuscript.

**Funding:** This research was supported by the 2022 Shandong Social Science Foundation Project "Research on teaching management innovation in rural primary schools in the post-pandemic era" chaired by I-Hua Chen (Project No.: 22CJYJ16).

**Institutional Review Board Statement:** The study was conducted in accordance with the Declaration of Helsinki and approved by the Institutional Review Board of Jiangxi Psychological Consultant Association (IRB ref of teachers: JXSXL2020-J013).

**Informed Consent Statement:** Informed consent was obtained from all subjects involved in the study.

**Data Availability Statement:** The data presented in this study are available on request from the corresponding author. The data are not publicly available due to the restriction by the institutional review board.

**Acknowledgments:** The authors would like to thank all the teachers who participated in this study.

**Conflicts of Interest:** The authors declare that the research was conducted in the absence of any commercial or financial relationships that could be construed as a potential conflict of interest.

## Appendix A

In Table A1, diagonal elements in bold are the square root of the averaged variance extracted. When these values were higher than the inter-latent factors' correlations (off-diagonal elements), the discriminant validity was a support for the respective latent variable.

**Table A1.** Discriminant validity of latent variables.

|  | PTSD | PD | IGD |
|---|---|---|---|
| PTSD | **0.865** |  |  |
| PD | 0.608 | **0.789** |  |
| IGD | 0.516 | 0.508 | **0.855** |

Notes. Sleep problems were measured by a single item that caused averaged variance extracted (AVE) not to be calculated, so it is not shown in the above list.

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
