# Peer review of "The Association between Post-Traumatic Stress Disorder and Psychological Distress among Primary School and Middle School Teachers during the COVID-19 Epidemic: A Moderated Mediation Analysis"

_sustainability, doi:10.3390/su141912128_

Round 1
Reviewer 1 Report
This study evaluated the mediating role of sleep problem and the moderating role of internet use on the relationship between traumatic experience (which is the intended independent variable) and mental health outcomes.
The study rationale, design and results are indeed very clearly presented. However, I have a critical concern on the measure of traumatic exposure. The current measure is the PCL-5 which reflects the severity of PTSD symptoms instead of exposure to traumatic experience. The measure has no surprise to have high correlation with other mood measures, that includes DASS-21. However the outcome of PCL-5 should be a psychological reaction, which is also the mental health outcomes, instead of being regarded as an independent variable conceptually. Wonder if authors have any other measures that really explore the level of exposure to traumatic experience?
Reviewer 2 Report
I have read with great interest the manuscript entitled: The Association Between Post-Traumatic Stress Disorder and Psychological Distress Among Primary School and Middle School Teachers During the COVID-19 Epidemic: A Moderated Mediation Analysis. In this study, Jia-Xin Liang et al aimed to verify the relationship between PTSD and psychological distress, and to explore the possible mediator of sleep problems in the relationship between PTSD and psychological distress, as well as the moderator of internet gaming disorders (IGD) in the relationship between sleep problems and psychological distress. The participants were 11,014 teachers who participated in the investigation from Jiangxi, Sichuan, and Shandong between May 25 and June 30, 2020, during the COVID-19 pandemic. The manuscript is of good impact, well conceptualized, and clearly written. The introduction provides sufficient background and includes all relevant references. The methods are sufficiently described. The results are clearly presented. The conclusions supported by the evidence presented. All the cited references are relevant to the research. The authors should explain the abbreviation: COVID-19, SARS, DSM-5 in the manuscript. The authors should shorten their abstract. The authors should change sections 3. Data Analyses into subsections 2.3 of Materials and Methods. I think that the topic of this manuscript is important and is suitable for Sustainability.
Thank you
Reviewer 3 Report
The article presents specific consequences of the COVID-19 pandemic on psychological state of schoolteachers. The aim is laudable and more research like this should be done. I appreciate the impressive subject sample, over 11000 people.
The hypotheses are well supported by the literature and are plausible.
The model and the statistical analysis are well presented.
The discussion section is support by secondary literature and has added the insights from present research.
With a sample that large, the statistic output is very robust.
Comments and suggestions:
1. Page 4, fig 1, there is not written H4. It is explained in the capitation, but it is not contained in the drawing.
Pg 5, section 2.2.3, it says four-point Liker, but it explains from 1 to 5.
